# The Notorious Triumvirate in Pediatric Health: Air Pollution, Respiratory Allergy, and Infection

**DOI:** 10.3390/children10061067

**Published:** 2023-06-15

**Authors:** Anang Endaryanto, Andy Darma, Tonny Sundjaya, Bertri Maulidya Masita, Ray Wagiu Basrowi

**Affiliations:** 1Faculty of Medicine, Universitas Airlangga, Surabaya 60132, Indonesia; aendaryanto.ae@gmail.com (A.E.); andy.darma@fk.unair.ac.id (A.D.); 2Medical and Science Affairs Division, Danone Specialized Nutrition Indonesia, Jakarta 12940, Indonesia; tonny.sundjaya@danone.com (T.S.); bertri.masita@danone.com (B.M.M.)

**Keywords:** air pollution, immune system, respiratory allergy, respiratory infection, pediatric health

## Abstract

A plausible association is suspected among air pollution, respiratory allergic disorder, and infection. These three factors could cause uncontrollable chronic inflammation in the airway tract, creating a negative impact on the physiology of the respiratory system. This review aims to understand the underlying pathophysiology in explaining the association among air pollution, respiratory allergy, and infection in the pediatric population and to capture the public’s attention regarding the interaction among these three factors, as they synergistically reduce the health status of children living in polluted countries globally, including Indonesia.

## 1. Acknowledging the Health Impact of Air Pollution among the Pediatric Population

The World Health Organization (WHO) [1] reported that 93% of children worldwide are exposed daily to unacceptable levels of particulate matter with a diameter of 2 × 5 μm or smaller (PM_2.5_). In the Southeast Asian region, 99% of children under five years old living in low- and middle-income countries were exposed to PM_2.5_ levels higher than those set forth in the WHO air quality guidelines. In 2016, approximately 600,000 deaths among children under 15 years old worldwide were attributed to air pollution; more than 90% of those were children under five years old. The Institute for Health Metrics and Evaluation [2] estimated that for Indonesia, 28.14 deaths and 2497.31 disability-adjusted life years (DALYs) per 100,000 population in 2019 were linked to air pollutant exposure, causing the third largest risk factors for mortality and morbidity among children under five years old. In the under-five pediatric population, respiratory infection was the second cause of lost DALYs linked to air pollution after neonatal disorders (722.42 DALYs per 100,000 population). Notably, 28.9% of the DALYs lost to air pollution were due to respiratory infections.

Air pollutants comprise two general components, i.e., particulate matters (PM) and gaseous components. Conventionally, PMs are thought to be the main cause of air-pollution-related health issues. However, studies showed that gaseous pollutants (e.g., NO_2_ and SO_2_) also affected health, as discussed subsequently. PM is usually described by its aerodynamic equivalent diameter, whereby it is categorized as (i) particulate matters larger than 10 µm, (ii) particulate matters with a diameter of 10 μm or smaller (PM_10_), (iii) PM_2.5_, and (iv) ultrafine PM (UFPM). While PM larger than 10 um is commonly filtered in the nose and upper airways, PM_10_ is usually deposited in the more proximal airways. Importantly, PM_2.5_ and UFPM could reach terminal bronchioles and alveoli [3].

Children are more vulnerable to air pollution than adults because of biological, behavioral, and environmental factors. Physiologically, since their organs, including the brain and lungs, are in the maturation phase, their bodies are more vulnerable to inflammation and other damage caused by air pollution. This risk is even higher during infancy because the faster breathing rates expose them to higher levels of air pollution. Children also tend to live closer to the ground, where some pollutants settle. During their early years, they are exposed more to household air pollution. Conversely, in their older years, they spend more time outdoors, inhaling more ambient air pollution [1].

## 2. Investigating the Influence of Air Pollution on Respiratory Allergic Disorder

Air pollution could activate the immune system through three pathways, i.e., Toll-Like Receptor (TLR), Reactive Oxygen Species (ROS), and Poly-Aromatic Hydrocarbon (PAH)-sensing pathways. First, evolutionarily designed for detecting pathogen-associated molecular patterns of microbes, the TLR pathways (particularly the TLR4 pathway) could be activated by PM, as in vitro and animal studies [4,5,6,7] have shown. The TLR pathways could also be activated by direct host cellular damage, subsequently producing alternative agonists to TLR ligands, such as oxidized phospolipids and nucleic acids [4]. Second, PM, through components of heavy metals and organic compounds, could generate ROS, resulting in oxidative stress. In addition, ozone (O_3_) and NO_2_ are two gaseous components that could cause oxidative stress in the exposed respiratory tract [8,9]. Oxidative stress can trigger ROS-activated inflammatory pathways and directly damage the respiratory epithelium [10]. Third, organic PAH could induce oxidative stress through a specific pathway, called the aryl hydrocarbon receptor (AhR). Activated AhR causes nuclear translocation of xenobiotic-metabolizing enzymes (i.e., CYP1A1 and CYP1B1), thus increasing the concentration of cytotoxic and genotoxic products [11]. These would trigger the cascade of airway inflammation through pro-inflammatory cytokines, chemokines, and other signaling molecules, including IL-6, CXCL8, and GM-CSF.

The principle of a healthy immune system distinguishes harmless self and dangerous non-self antigens. The harmless antigens comprise their own cells, beneficial microbiota, and innocuous environmental antigens, while dangerous antigens include infectious microbes and neoplastic cells. Hypersensitivity diseases (also known as allergic disorders) occur due to the inability of the immune system to differentiate harmless self from dangerous non-self antigens, resulting in the harmful T_H_2-cell-mediated immune response among allergic patients [12,13]. Although allergic respiratory disease is not well characterized, its common definition refers to pathologic and symptomatic acute or chronic hypersensitivity reactions within the respiratory tract upon exposure to a specific allergy-inducing antigen (i.e., allergen), in which the condition is exacerbated by previous immunological sensitization and production of immunoglobulin E against the allergen [14,15]. The respiratory allergy could manifest as allergic rhinoconjunctivitis, allergic rhinitis, or asthma [14,15].

Although air pollution’s ability to directly cause respiratory allergy [16] is elusive, air pollution triggers pro-inflammatory cytokines through the three pathways mentioned above, exacerbating the allergic inflammatory reaction in the respiratory tract. Following epidemiological evidence suggests that air pollution could worsen asthma symptoms and increase the risk of developing asthma. A meta-analysis by Han et al. [17] showed that traffic-related air pollution (TRAP) increased the risk of asthma development in children, reporting the odds ratio (OR) of PM_2.5_ = 1.07 (95% confidence interval (CI): 1.00–1.13), the OR of NO_2_ = 1.11 (95% CI: 1.06–1.17), the OR of Benzene = 1.21 (95% CI: 1.13–1.29), and the OR of TVOC (total volatile organic pollutants) = 1.06 (95% CI: 1.03–1.10). Yan et al. [18] even reported that maternal exposure to PM_2.5_ during pregnancy could increase the risk of childhood asthma and wheezing (OR = 1.06; 95% CI: 1.02–1.11; per 5 μg/m^3^ increase). Furthermore, asthma exacerbation and its outcomes were linked to air pollution. TRAP and the individual pollutants (PM_2.5_, NO_2_, and SO_2_) have been linked to the inability to control asthma, medication cost increase, hospitalization duration, and mortality [19]. Air pollution could also affect allergic rhinitis, in which PM_2.5_ (OR = 1.09; 95% CI: 1.01–1.17; per 10 μg/m^3^ increase) had a slightly larger impact than PM_10_ (OR = 1.06; 95% CI: 1.02–1.11; per 10 µg/m^3^ increase) [20]. Zou et al. [12] also supported those findings by demonstrating that exposure to NO_2_ (OR = 1.138; 95% CI: 1.05–1.23) and SO_2_ (OR = 1.085; 95% CI: 1.01–1.16) increased the risk of childhood allergic rhinitis.

Another proposed pathway on how air pollution affects allergic disorders is through an interaction between air pollutants and environmental allergens. For example, a significant CO_2_ increase within the urban environment induced ragweed (*Ambrosia* sp.) to grow faster and bloom earlier, increasing its pollen [21] production. An increased concentration of CO_2_ was shown to increase the production of spores of several types of molds [22]. In addition, the allergenicity of birch pollen increased due to the higher concentration of O_3_ [23]. However, detailed biomechanical studies are required to understand the pathophysiology of how air pollution could influence or even incite hypersensitivity diseases within the respiratory system and how the air pollutants would interact with environmental allergens in increasing the severity of respiratory allergic disorders [24].

## 3. Studying the Influence of Air Pollution on Respiratory Infection

The epithelium in the respiratory system consists of ciliated epithelial and mucus-producing goblet cells. Generally, air pollution could increase the risk of respiratory infection by impairing the immune responses in the respiratory tract. First, a murine study demonstrated that inhaled diesel engine emissions reduced the airway clearance of *Pseudomonas aeruginosa* and aggravated lung histopathology during the bacterial infection [25]. A dysregulated innate immune response could contribute to impairment of the bacterial clearance in air-pollutant-induced airway epithelial dysfunction. [25,26]. Lung injury was reported to decrease alveolar macrophage internalization of bound bacteria and, therefore, lowered the absolute numbers of bacterial death [27]. Of note, air pollution could induce microbial dysbiosis within the lung [26]. The dysbiosis in the lung microbiome existed in various respiratory inflammatory diseases (e.g., chronic obstructive pulmonary disease), but whether it was the cause or result of those diseases [28] remains elusive. Second, chronic exposure to diesel exhaust particles resulted in the cellular toxicity of monocyte-derived macrophages, causing apoptosis and response impairments toward pathogenic stimuli [29]. Third, exposure to air pollutants also generated oxidative stress within respiratory epithelial cells, increasing the susceptibility to viral infection (e.g., influenza virus) as more viruses could attach and enter the stressed epithelial cells [30]. Fourth, exposure to air pollutants in the form of carbon black shifted the immune response from T_H_1-cell-mediated immunity (essential for bacterial and viral clearance) to T_H_2-cell-mediated immunity, hence, facilitating the occurrence of allergic inflammation. It has been shown in mice that exposure to ultrafine carbon black particles before respiratory syncytial virus infection increased the production of interleukin 13 (a T_H_2 cytokine) but decreased the production of interferon gamma (an antiviral cytokine) [31].

Furthermore, Glencross et al. [3] described two pathways by which air pollutants caused diseases: dysregulation of immune tolerance and dysregulation of antibacterial and antiviral immune responses (Table 1).

A short-term linear association was observed between air pollutants and pediatric hospital admissions et causa pneumonia. The excess risks for an increase of 10 µg/m^3^ of PM_10_ and of PM_2.5_ were 1.5% and 1.8%, respectively. Increments of SO_2_, O_3_, and NO_2_ per 10 parts per billion caused 2.9%, 1.7%, and 1.4% excess risks of contracting pediatric pneumonia, respectively [32]. Of note, no short-term association was found with CO. Intriguingly, a disproportionately larger risk of acute lower respiratory infections (i.e., bronchitis and pneumonia) were found in the pediatric population when the exposure to air pollutants was sub-chronic to chronic. Furthermore, an increase of 10 µg/m^3^ of PM_2.5_ was shown to heighten the excess risk to 12% [33]. Importantly, outcomes of tuberculosis were strongly associated with air pollution as well. However, PM_2.5_ were the only pollutant frequently associated with tuberculosis because studies on PM_10_, SO_2_, and NO_2_ showed inconsistent results [34]. Emerging in early 2020, the Coronavirus Disease 2019 (COVID-19), a primarily infectious disease in the respiratory system caused by the severe acute respiratory syndrome coronavirus 2 (SARS-CoV-2), has remained a pandemic these past three years. The persistence of this disease is partially due to the continuous presence of new variants of SARS-CoV-2 [35]. Thus, it would be interesting to investigate the impact of air pollution on the incidence and severity of COVID-19 worldwide. Although a range of epidemiological and animal studies suggested a link between air pollution and COVID-19, more investigational studies are required to confirm this association [36].

Regarding the acute upper respiratory infection, the evidence pointed out that ambient air pollution was associated with otitis media in children. Middle ear exposure to air pollution could increase mucin and inflammatory cytokine expression, leading to the blockade of the Eustachian tube. It was hypothesized that Eustachian tube obstruction promoted microbial migration from the nasopharynx to the middle ear and microbial growth within the middle ear, resulting in otitis media. However, since only NO_2_ exhibited a significant correlation with the otitis media [37], this hypothesis requires further investigation and refinement.

Taken together, these population-based studies suggest a plausible causative relationship between air pollution and respiratory infections. However, due to the complexity of air pollutant mixtures and the confounding factors of population-based studies, the pathophysiology of how air pollutants affect respiratory infection can be established only through in vitro and animal studies [38].

## 4. Mapping the Relationship among Air Pollution, Allergic Disorder, and Infection

As the incidence and severity of various respiratory allergy disorders and infections among children appeared to be high in polluted countries, investigating any association among these three factors is interesting. For example, are there any causative relationships among air pollutants, respiratory allergic disorder, and infection? As discussed, the causative role of air pollution for respiratory allergic disorder or infection has been suggested but has not yet been established. Therefore, this section discusses the plausible association between respiratory allergic disorder and infection, particularly among the pediatric population.

The predominant immune response in allergic disorders is T_H_2-cell-mediated immunity, in which this immune response is evolutionarily conserved for parasite clearance [13]. This skewing hinders the host’s ability to clear bacterial and viral infections; therefore, respiratory allergic disorders could predispose the hosts to contract certain respiratory infections. This notion was partially supported by a published study, reporting that rhinovirus’ replication in asthmatic bronchial epithelial cells resulted in a higher yield of the virus than in healthy bronchial epithelial cells [39]. Furthermore, a recent clinical trial demonstrated that providing house dust mite sublingual allergen immunotherapy for twenty asthmatic patients improved bronchial epithelial resistance against viral infection [40], suggesting that attenuating allergic disorders could protect against certain respiratory infections.

Conversely, it has been noted that among some children, certain respiratory infections in early life, as indicated by wheezing episodes, could mark the beginning of asthma [41]. Although the exact pathophysiology has not been established, two wheezing-induced viruses (i.e., respiratory syncytial virus and rhinoviruses) have been associated with asthma inception [42]. Furthermore, it was reported that an infection of atypical bacteria *Chlamydia pneumoniae* could initiate asthma in previously non-asthmatic patients [41]. The notion that certain infections may precede allergic disorders was supported by observational studies in Finland and Sweden, suggesting that the occurrence of respiratory infections in the past 12 months was the determinant for the onset of adult asthma [43] and that a history of severe respiratory syncytial-virus-mediated bronchiolitis during infancy was associated with an increased risk of developing asthma in early adulthood [44]. However, although asthma is usually mentioned as a severe allergic disorder in the respiratory system, it has heterogenous phenotypes comprising allergic and non-allergic asthma, in which the incidences of allergic and non-allergic asthma appeared to peak in early childhood and late adulthood, respectively [45,46]. Thus, the pending question is whether the reported respiratory infections precede allergic asthma, non-allergic asthma, or both. Next, certain viral infections were associated with asthma exacerbations in older children and adults, in which it was hypothesized that respiratory viruses could interact with allergens to exacerbate asthma [45,47]. This hypothesis was supported by a finding that upper respiratory tract infection was associated with 80–85% of asthma exacerbations among school-age children [48].

Taken together, there is no definite proof yet of a causal relationship among air pollution, respiratory allergic disorder, and infection in the pediatric population. Since elucidating the pathophysiology linking air pollution to allergic disorder and infection is challenging, the attention should be focused instead on the aggregated negative impact of these three factors on health status because they could co-exist in patients and synergistically could cause unwanted chronic inflammation within the respiratory system. Indeed, air pollution, respiratory allergy, and respiratory infection commonly co-exist and, hence, are likely to be associated. A birth cohort study with ~4000 subjects in the Netherlands supported this notion by reporting that a positive association exists among traffic-related air pollution (NO_2_, PM_2.5_, and “soot”), asthma occurrence, other allergic disorders, as well as respiratory infection at 2 and 4 years of age [49,50]. An acknowledgment that air pollution could synergize with allergic disorders and respiratory infection in reducing the functionality of the respiratory system and that this is a global issue should, arguably, facilitate concerted efforts by scientists, physicians, and public health officers to address this imminent issue adequately.

## 5. Conclusions

Although the epidemiological evidence on how air pollutants affect allergic respiratory disorder and infection is abundant and indicative, elucidating the biomechanisms linking these three factors is very challenging. Nonetheless, the inability to explain the relationship among air pollution, respiratory allergy, and infection should not hinder the effort to highlight the gravity of this health issue. Instead, all relevant parties should acknowledge that in the era of global industrialization, air pollution could worsen the morbidity and mortality caused by respiratory allergic disorders and respiratory infection globally, including in Indonesia.

## Figures and Tables

**Table 1 children-10-01067-t001:** Dysregulation of immune tolerance and antimicrobial responses due to air pollution [3].

Dysregulation of Immune Tolerance	Dysregulation of Antimicrobial Immunity
Stimulation of production of pro-inflammatory cytokines and leucocyte-attracting chemokines by epithelial cells and macrophages.	Overladen macrophages with diminished phagocytic capacity.
Adjuvant action of PM increased APC maturation and antigen expression.	NO_2_ increased epithelial expression of ICAM-1 (receptor for respiratory viruses).
Suppression of pro-inflammatory cytokines (such as IL-6) by regulatory T cells.	Dysregulation of IFN-γ production by T cells.
Protein oxidation leading to formation of neo-antigens.	Development of T_H_2-biased inflammation that could not control microbial infection.

APC, antigen-presenting cells; ICAM-1, intercellular adhesion molecule 1; IFN-γ, interferon gamma; IL-6, interleukin 6; PM, particulate matters; T_H_2, T helper 2.

## Data Availability

Not applicable.

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
