# Peer review of "The Notorious Triumvirate in Pediatric Health: Air Pollution, Respiratory Allergy, and Infection"

_children, 2023, doi:10.3390/children10061067_

Round 1

Reviewer 1 Report

Endaryanto et al. have conducted a narrative review of pediatric respiratory health, allergy and air pollution exposure.

The field is of interest and the areas covered are notable, though it is a shame only 42 papers are covered in this review, significantly limiting the scope and interest to the general audience. The term respiratory allergy is used throughout the manuscript, including the title but is poorly defined and not clear what exactly is being referred to here. A more precise definition of the terminology would be beneficial.    eg line 127, what does exacerbation of certain viral infections mean clinically? What virus? Frequent imprecise statements throughout make the manuscript very difficult to understand and limit its interest and relevance. 

The standard of written English is low and prevents a comprehensive review at this time. At many points, it is not clear to this reviewer if the statement is a factual error or confounded by poor grammar. At other points, the descriptions of published works are correct but too vague to give the reader a clear understanding of the research findings or limitations.  I would recommend the authors seek editing and assistance as the manuscript requires significant work prior to being suitable for peer review or to publish.

Author Response

Dear Reviewer,

Thank you for your robust and valuable reviews, please find attached the revisions we have made based on your suggestions.

Regards,

Ray

Reviewer 2 Report

This is an interesting, a bit superficial, short description of a complicated relationship between air pollution, allergy, and infection.

My comments:

everywhere, where OR is given, it should be accompanied by CI!

To be corrected:

line 9: proof of the linearity of the relationship (in my opinion it is NOT linear)

line 60: I am doubting about the "similarity of PM to microbial molecules, such as lipopolysaccharides". Some references?

lines 115-116: it is not true that LUNG injury is compromising the microbial clearance in the airway

small corrections are needed

Author Response

Dear Reviewer,

Thank you for your robust and valuable reviews and inputs, and please find attached the revisions we have made based on your suggestions.

Regards,

Ray

Round 2

Reviewer 1 Report

.

Reviewer 2 Report

Thank you for the corrections